# SARS-CoV-2 Infection in Companion Animals: Prospective Serological Survey and Risk Factor Analysis in France

**DOI:** 10.3390/v14061178

**Published:** 2022-05-28

**Authors:** Pierre Bessière, Timothée Vergne, Matéo Battini, Jessie Brun, Julien Averso, Etienne Joly, Jean-Luc Guérin, Marie-Christine Cadiergues

**Affiliations:** 1IHAP, Université de Toulouse, INRAE, ENVT, 31076 Toulouse, France; pierre.bessiere@envt.fr (P.B.); timothee.vergne@envt.fr (T.V.); mateo.battini_17@envt.fr (M.B.); jean-luc.guerin@envt.fr (J.-L.G.); 2Small Animal Clinic, Université de Toulouse, ENVT, 31076 Toulouse, France; jessie.brun@envt.fr (J.B.); averso.julien@gmail.com (J.A.); 3Institute of Pharmacology and Structural Biology (IPBS), Université de Toulouse, CNRS, 31400 Toulouse, France; etienne.joly@ipbs.fr; 4INFINITy, Université de Toulouse, INSERM, CNRS, UT3, ENVT, 31059 Toulouse, France

**Keywords:** SARS-CoV-2, dog, cat, serology, epidemiology, survey, risk factors

## Abstract

Severe acute respiratory syndrome coronavirus-2 (SARS-CoV-2), responsible for COVID-19 in people, has been detected in companion animals on rare occasions. A limited number of large-scale studies have investigated the exposure of companion animals to SARS-CoV-2. The objective of this prospective study was to estimate seroprevalence in privately owned dogs and cats presented in veterinary clinics in different French regions and to test the hypothesis that the occurrence of an episode of COVID-19 in the household and close contact with the owner would increase the chances of the animals being seropositive. One hundred and sixty-five dogs and 143 cats were blood-sampled between March 2020 and December 2021. Neutralizing SARS-CoV-2 antibodies were detected in 8.4% of cats (12/143) and 5.4% of dogs (9/165). Seven animals (three dogs and four cats) were seropositive in the absence of an episode of COVID-19 in the household. Despite not being statistically significant (chi-square test, *p*-value = 0.55), our data may suggest that the occurrence of an episode of COVID-19 in the household could increase the risk of animal seropositivity (odds ratio = 1.38; 95% confidence interval = 0.55–3.77). This survey indirectly shows that SARS-CoV-2 circulates in canine and feline populations, but its circulation appears to be too low for pets to act as a significant viral reservoir.

## 1. Introduction

Severe acute respiratory syndrome coronavirus 2 (SARS-CoV-2) emerged in 2019 in China and rapidly spread worldwide [1]. While the primary virus host is humans, several spill-over events have been reported that, in some cases, have resulted in sustained animal-to-animal transmission [2]. Among domestic animals, cats and dogs are susceptible hosts to SARS-CoV-2 infection. With viral shedding being frequently weak and transient in these species [3,4], detection of anti-SARS-CoV-2 antibodies is the best way to monitor viral circulation. A limited number of large-scale studies have investigated the exposure of companion animals to SARS-CoV-2 [5,6,7,8,9,10,11]. Our study contributes to this evidence by estimating seroprevalence in cats and dogs in France and testing the hypothesis that the occurrence of an episode of COVID-19 in the household and close contact with the owner increase the chances of the animals being seropositive. A secondary objective was to compare two serological diagnostic methods: an enzyme-linked immunosorbent assay (ELISA) and a hemagglutination-based assay.

## 2. Materials and Methods

### 2.1. Ethics

The animal experiments were approved by the Sciences et Santé Animales (SSA–Ecole Nationale Vétérinaire Toulouse) N°115 Ethics Committee (Approval No. SSA_2020_010). Written consent from the pets’ owners was obtained prior to the beginning of the study.

### 2.2. Sample Collection

The survey was conducted prospectively during healthcare visits at veterinary clinics or veterinary university hospitals between March 2020 and December 2021 throughout metropolitan France. Information on each sampled animal was registered by the veterinarian. It included the signalment of the animal (species, breed, age, and gender), its lifestyle (indoor/outdoor and time spent outdoors), the occurrence of clinical signs suggestive of SARS-CoV-2 infection (anorexia, hyperthermia, lethargy, gingivitis, polyadenitis, diarrhea, stomatitis, bronchopneumonia, cough, and/or other), and the proximity between the pet and the owner (cuddling, playing, or frequent grooming). The information obtained from the owners included the occurrence of a COVID-19 episode in the household and, if any, the severity of the owner’s symptoms and the availability of a positive test result were recorded. The data were entered online on a dedicated platform allowing pseudonymization. Blood samples were taken by standard venepuncture. The serum was collected by centrifugation, and stored at −20 °C until analyses.

### 2.3. Hemagglutination Test

The hemagglutination test (HAT) was used as previously described [12]. The assay was carried out in 96-well plates, with 50 µL of IH4-RBD reagent (2 µg/mL), 1 µL of undiluted serum, and 1 µL of human red blood cells from an O negative donor diluted at 30% *v*/*v* in PBS. To detect xenoreactive sera, for every sample, a negative control was performed using PBS instead of the IH4-RBD reagent. Feline serum from a previous experiment was used as a positive control. Plates were incubated for 1 h at room temperature and briefly centrifuged. Then, they were tilted to detect the lack of teardrop formation indicative of a positive reaction.

### 2.4. ELISA

The serological status of the animals was assessed using a commercial SARS-CoV-2 N double antigen enzyme-linked immunosorbent assay (ELISA) kit in accordance with the manufacturer’s instructions (ID Screen^®^ SARS-CoV-2 Double Antigen Multi-species ELISA, ID.vet, Grabels, France).

### 2.5. Serum Neutralization Assay

ELISA and HAT positive or inconclusive sera were further tested using a virus serum neutralization assay, which was carried out using Vero-E6 cells, as described previously [3]. Briefly, samples were heat-inactivated for 30 min at 56 °C, serially diluted in PBS, and mixed with 100 TCID_50_ of SARS-CoV-2 (strain hCoV-19/France/OCC-IHAP-VIR12/2020), in a final volume of 100 µL. After being incubated for 1 h, virus serum mixtures were deposited onto cells. Cells were washed with PBS 1 h later and then incubated in infection media (DMEM complemented with 2% of heat-inactivated fetal bovine serum and 1% of penicillin-streptomycin) for 72 h at 37 °C, with 5% CO_2_. They were then screened for cytopathic effects. Serum neutralization titers were defined as the highest serum dilution inhibiting viral growth. Experiments were carried out in a biosafety level 3 facility at the National Veterinary School of Toulouse. 

### 2.6. Statistical Analysis

The univariable associations between the serological test results and all putative risk variables were tested using a Fisher test for all animals together and for cats and dogs separately. A multivariable logistic regression on the serological status was also performed using both the species and the previous occurrence of a COVID-19 episode in the household as explanatory variables, as these two variables were the main focus of the analysis. The other variables studied were the severity of the COVID-19 episode (asymptomatic, mild, or serious), whether the owners had performed a test (PCR or serological) to confirm their infection, and the nature of their relationship with their pet (close or distant).

## 3. Results

We collected 314 samples. After the exclusion of six cats (owing to an unfilled questionnaire), a total of 308 privately owned dogs (165) and cats (143) from different French regions (Figure 1) were included in the study. 

Sera were first screened using two techniques: ELISA and HAT. Sample positivity was then confirmed by a virus serum neutralization assay. Sera were considered positive only if the virus serum neutralization assay was positive. Neutralizing SARS-CoV-2 antibodies were detected in 12 cats (8.4%, 12/143) and 9 dogs (5.4%, 9/165).

While all ELISA positive sera were serum neutralization positive, seven feline samples (4.9%, 7/143) and seven canine samples (4.2%, 7/165) were HAT positive and ELISA negative, but serum neutralization positive. HAT thus appears more sensitive than ELISA, and possibly also more sensitive than serum neutralization because nine cat and 20 dog sera were HAT positive, but showed no viral neutralization activity in our assay. On the other hand, a sizeable number of sera (6.8%, 21/308), from 16 cats and five dogs, contained xenoreactive antibodies, causing hemagglutination in the HAT negative control, and were all found to be negative in the virus neutralization assay. Detailed information regarding all sera and tests results can be found in Appendix A. 

In the univariable analysis, none of the tested predictor variables were significantly associated with the animals’ serological status (Table 1). In the multivariable logistic regression, the odds ratio (OR) of being seropositive for cats, as compared with dogs, was estimated to be 1.51 (95% confidence interval: 0.61–3.84). Similarly, the OR of being seropositive for pets from a household with a previous COVID-19 episode, as compared with pets from a household without a previous COVID-19 episode, was estimated to be 1.38 (95% CI: 0.55–3.77).

## 4. Discussion

This study reveals that, although not systematic, the transmission of SARS-CoV-2 from humans to domestic carnivores is far from rare. Anti-SARS-CoV-2 antibodies were found in 8.4% and 5.4% of cat and dog samples, respectively. Interestingly, whether or not the owners had contracted COVID-19 was not statistically significantly associated with the serological status of the animals, in contrast to the results of a large-scale study [10]. However, the associated OR, adjusted for the species, suggests a potential association of practical interest (1.38; 95% CI: 0.55–3.77). Two hypotheses are thus possible: either the risk of contamination of pets outside the household is substantial, or, more realistically, the sample size was too small to reveal a significant difference.

The different seroprevalence studies performed in Europe reported variable positivity rates. While studies performed in Germany, the United Kingdom, and Croatia found positivity rates below 3% [9,13,14], an Italian study reported higher rates: 3.3% and 5.8% in cats and dogs, respectively [10]. Several factors may explain the differences: geographical location, the number of cases reported in humans, and the way the animals were included in the study, among others. As some coronaviruses are endemic in companion animals, one may wonder if infections prior to the emergence of SARS-CoV-2 could be the cause of false positives. As previously reported, there is no cross-reactivity between neutralizing antibodies induced by SARS-CoV-2 infection and those induced by canine and feline coronavirus infections [13].

Our study has several limitations. Firstly, the size of our cohort was limited—we collected about three hundred samples, while other studies have reached a thousand [6,9]. Secondly, the samples were collected over a long period of time, so we were unable to take the influence of the different variants of SARS-CoV-2 into account. Human-to-domestic animal SARS-CoV-2 transmission probably mirrors the rise in registered human cases, as previously suggested [9]. The more recent the sampling campaign, the more likely it is that the tests could be positive. Nevertheless, the majority of our positive samples (67%, 14/21) were taken in 2020, early in the pandemic. As a significant number of sera were collected in the south-west of the country (the Occitanie region), one might wonder to what extent this introduces a bias into our analysis, in addition to the fact that France has undergone several epidemic waves. Nevertheless, according to the database used to explore data on the epidemic in France (https://covidtracker.fr/covidexplorer/; accessed on 25 May 2022), the epidemiological situation in the Occitanie region was broadly similar to that of the rest of the country throughout the first waves.

We also tested the HAT designed by Townsend et al. [12], which is inexpensive, quick and easy to use; and, from our results, more sensitive than ELISA. Despite these advantages, we would not recommend its use for dog and cat samples because of the high proportion of sera containing xenoreactive antibodies (6.8%), making the test inconclusive.

In conclusion, SARS-CoV-2 has been shown to circulate in canine and feline populations, but its circulation appears to be too low for pets to act as a viral reservoir. Nevertheless, viral circulation needs to be monitored, as it cannot be excluded that variants (such as immune-escaping variants) may emerge in domestic carnivores.

## Figures and Tables

**Figure 1 viruses-14-01178-f001:**
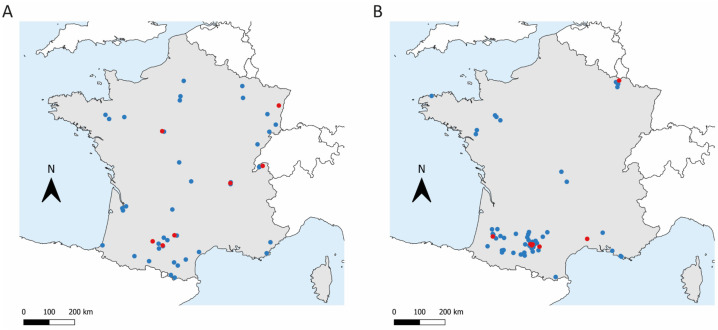
Spatial distribution of the feline (**A**) and canine (**B**) sample locations. Blue dots indicate samples that were negative for anti-SARS-CoV-2 antibodies. Red dots indicate samples that were positive.

**Table 1 viruses-14-01178-t001:** Detailed information on dogs’ and cats’ serological status and risk factors in the household, in the univariable analysis.

		Serological Status for All Pets		*p*-Value *	Serological Status for Dogs		*p*-Value **	Serological Status for Cats		*p*-Value **
		Positive	Negative		Positive	Negative		Positive	Negative	
Species (n = 308)	dog	9	156	0.42						
	cat	12	131							
Relationship pet–owner (n = 305)	close	16	199	0.73	5	95	0.74	11	104	0.69
	distant	5	85		4	61		1	24	
COVID-19 episode in the household (n = 308)	yes	14	165	0.55	6	77	0.50	8	88	1
	no	7	122		3	79		4	43	
Severity of the COVID-19 episode (n = 308)	serious	4	27	0.34 ***	2	8	0.11	2	19	1
	mild	10	138		4	69		6	69	
	none	7	122		3	79		4	43	
COVID-19 owner test (n = 292)	positive	11	121	0.40 *	6	65	0.41	5	56	0.71
	negative	3	29		0	9		3	20	
	no test	6	122		3	82		3	40	

* Chi-square test. ** Fisher’s exact test. *** A Fisher’s exact test was used instead of the chi-squared test because at least one of the expected numbers was less than 5.

## Data Availability

The data presented in this study are available on request from the corresponding author. The data are not publicly available because of the need to maintain patient confidentiality.

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
