# Peer review of "SARS-CoV-2 Infection in Companion Animals: Prospective Serological Survey and Risk Factor Analysis in France"

_viruses, 2022, doi:10.3390/v14061178_

Round 1
Reviewer 1 Report
The manuscript entitled “SARS-CoV-2 Infection in Companion Animals: Prospective Se rological Survey and Risk Factors Analysis in France” was reviewed.
This is an interesting study on the seroprevalence of antibodies against SARS-CoV2 in dogs and cats. Therefore, the title should be more precise and refer to exposure of companion animals to SARS-CoV-2 as absence of antibodies cannot exclude infection.
Line 20: Authors have added dogs and cats which is impossible because the sum cannot be defined. The number of each species sampled should be presented separately.
Lines 26-27: In order to assess the importance of dogs and cats as reservoir of the virus, viral detection instead of antibodies would have been more appropriate. In the present study, the presence of antibodies is an indication of the circulation of the virus, but the presence of the virus is indirectly assessed.
Analytical methodology followed to detect antibodies is most appropriate, assures the reliability of results and excellent presented. In general, the epidemiological study and the objective of this study as it is stated at the end of the introduction part is perfect. However, due to the limited number of samples taken, given the population of dogs and cats in France, the most important findings of the study are the risk factors analyzed. Statistical analysis is also most appropriate.
Another important issue that is not highlighted as much as it should have been done is the use of three different methods to detect antibodies and the comparative interpretation of the results. I would recommend to the author to add a secondary objective to their study regarding the different methods employed.
I feel that the work that have been done is rather underestimated as a lot of interesting findings can be presented not in the supplementary file but in the text.
Reviewer 2 Report
Bessiere et al performed the prospective serological survey of SARS-CoV-2 infection in companion animals. They found that SARS-CoV-2 circulates in canine and feline population, whereas the prevalence of SARS-CoV-2 sero-positivity is low. The study provides the epidemiological evidence how companion animals participate the COVID-19 spread.
- line 29 : which strain of SARS-CoV-2 was used?
- How to determine the serum neutralization titers? Is this plaque assay or TCID assay? And please provide the serum neutralization titers in main figure.
Reviewer 3 Report
The article “SARS-CoV-2 Infection in Companion Animals: Prospective Serological Survey and Risk Factors Analysis in France” by Pierre Bessière and co-authors describe a small seroprevalence study regarding SARS-CoV-2 neutralizing antibodies in cats and dogs.
In the discussion, it is mentioned that various factors such as the number of reported human cases or the geographic location of sampling need to be considered for comparison of different studies. Only the COVID-19 status of pet owners is reported in the results. It would increase the power of the study if it could be shown how many human cases occur in the sample areas during the coverage period. Since most samples were collected in the south of France (according to Figure 1 in the Toulouse region), there could be a bias regarding regional results. The authors themselves discuss that the study has limitations especially the size of the cohort they have analyzed. For publication the scientific significance should be enhanced by adding data about the incidence of COVID-19 during the study in the human population and the data collected should be correlated to it.
The conclusion that domestic carnivores are very unlikely to be a reservoir for SARS-CoV-2 is not novel, as other studies have already shown that transmission between cats is low and between dogs is not relevant at all. The risk analysis mentioned in the title is limited to a correlation of serologically positive animals in households with owners suffering from COVID-19. The mentioned incidences should in any case also be included in the risk analysis. From the methods part it is not clear which putative risk variables have been tested besides the occurrence of the disease in the households. Therefore it is recommended, to clarify the variables more or to change the title, as the title makes you expect more.
